

# Effects of preservation method on canine (*Canis lupus familiaris*) fecal microbiota

Katti R. Horng[1,2], Holly H. Ganz[3,4], Jonathan A. Eisen[3] and Stanley L. Marks[5]

[1] Department of Medical Microbiology and Immunology, University of California, Davis, Davis, CA, United States of America
[2] William R. Pritchard Veterinary Medical Teaching Hospital, School of Veterinary Medicine, University of California, Davis, Davis, CA, United States of America
[3] Department of Evolution and Ecology, University of California, Davis, Davis, CA, United States of America
[4] AnimalBiome, Inc., Oakland, CA, United States of America
[5] Department of Medicine and Epidemiology, School of Veterinary Medicine, University of California, Davis, Davis, CA, United States of America

## ABSTRACT

Studies involving gut microbiome analysis play an increasing role in the evaluation of health and disease in humans and animals alike. Fecal sampling methods for DNA preservation in laboratory, clinical, and field settings can greatly influence inferences of microbial composition and diversity, but are often inconsistent and under-investigated between studies. Many laboratories have utilized either temperature control or preservation buffers for optimization of DNA preservation, but few studies have evaluated the effects of combining both methods to preserve fecal microbiota. To determine the optimal method for fecal DNA preservation, we collected fecal samples from one canine donor and stored aliquots in RNAlater, 70% ethanol, 50:50 glycerol:PBS, or without buffer at 25 °C, 4 °C, and −80 °C. Fecal DNA was extracted, quantified, and 16S rRNA gene analysis performed on Days 0, 7, 14, and 56 to evaluate changes in DNA concentration, purity, and bacterial diversity and composition over time. We detected overall effects on bacterial community of storage buffer ($F$-value = 6.87, $DF = 3$, $P < 0.001$), storage temperature ($F$-value=1.77, $DF = 3$, $P = 0.037$), and duration of sample storage ($F$-value = 3.68, $DF = 3$, $P < 0.001$). Changes in bacterial composition were observed in samples stored in −80 °C without buffer, a commonly used method for fecal DNA storage, suggesting that simply freezing samples may be suboptimal for bacterial analysis. Fecal preservation with 70% ethanol and RNAlater closely resembled that of fresh samples, though RNAlater yielded significantly lower DNA concentrations ($DF = 8.57$, $P < 0.001$). Although bacterial composition varied with temperature and buffer storage, 70% ethanol was the best method for preserving bacterial DNA in canine feces, yielding the highest DNA concentration and minimal changes in bacterial diversity and composition. The differences observed between samples highlight the need to consider optimized post-collection methods in microbiome research.

Corresponding author
Katti R. Horng, krhorng@ucdavis.edu

## INTRODUCTION

The intestinal microbiota is comprised of trillions of bacteria that contribute to nutrition, digestion, immune defense, and various disease processes (*Petersen & Round, 2014*; *West et al., 2015*). However, it is estimated that 60–70% of resident gut bacteria cannot be discerned using culture-dependent methods (*Hayashi, Sakamoto & Benno, 2002*). Microbiota research involving culture-independent techniques such as DNA sequencing have made headway in the past decade with recent technological advances in next-generation sequencing and bioinformatics, providing more accurate taxonomic analysis and reproducibility between studies (*Wu et al., 2010*). The human gut is the perhaps the most well-known and well-studied microbiome, but there is increasing interest in other biological sites in humans, domestic animals, and wildlife in an effort to study the interaction between the host and its environment (*Paulino et al., 2006*; *Verhulst et al., 2011*; *Weese, 2013*; *Fliegerova et al., 2014*; *Lowrey et al., 2015*). These studies have paved the way for linking distinct bacterial communities within and between individuals. Recently, these efforts have greatly contributed to molecular fingerprinting and DNA identification in forensic science (*Fierer et al., 2010*; *Tims et al., 2010*). Despite the technological advances in noninvasive genetics (*Beja-Pereira et al., 2009*), there remains a lack of standardized methods for sample collection, bacterial preservation, and DNA extraction (*Yuan et al., 2012*; *Kennedy et al., 2014*; *Gorzelak et al., 2015*). Many studies that have evaluated bacterial preservation and DNA extraction showed variability in processing samples, leading to significant over- or underrepresentation of bacterial populations. For example, fecal storage at room temperature decreases the relative abundance of Firmicutes and increases Bacteroidetes, whereas storage in freezing conditions introduces the opposite effect (*Bahl, Bergstrom & Licht, 2012*; *Choo, Leong & Rogers, 2015*). Furthermore, the use of fecal swabs in human patients led to an overestimation of Enterobacteriaceae and Ruminococcaceae bacterial families (*Tedjo et al., 2015*). This highlights the need for more comprehensive evaluation of current techniques for optimal fecal DNA storage and isolation.

For many research laboratories, it is a challenge to minimize exposure of samples to the environment and minimize time between sample collection and DNA extraction (*Hale et al., 2016*). Preserving DNA through deactivation of nucleases, removal of cations or lowering temperature becomes crucial to inhibit enzymes that degrade DNA. Most studies have utilized freezing conditions (*Carroll et al., 2012*; *Wu et al., 2010*) or the use of preservatives such as ethanol (*Murphy et al., 2002*; *Bressan et al., 2014*; *Huang et al., 2016*), RNAlater (*Nechvatal et al., 2008*; *Sorensen et al., 2016*), and glycerol:PBS (*McKain et al., 2013*; *Fliegerova et al., 2014*) to preserve bacterial DNA before extraction. However, few studies have examined the combination of chemical buffers and temperature reduction to preserve DNA and optimize bacterial analysis. We designed a longitudinal study to evaluate the effects of temperature (room temperature at 25 °C, refrigeration at 4 °C, freezing at −80 °C) as well as immersion in 70% ethanol, RNAlater, and 50:50 glycerol:PBS on fecal DNA preservation over the course of a 56-day storage period.
## METHODS

### Stool collection and storage

One fresh stool sample from a healthy canine donor was collected at the UC Davis Veterinary Medical Teaching Hospital. The donor, a male Labrador retriever, was on a consistent commercial diet and had access to both indoors and outdoors. The donor was evaluated annually for its entire life by a veterinarian and had not been exposed to oral antibiotics 36 months prior to fecal collection. Fecal sampling occurred immediately after defecation. The sample was then homogenized and processed (within 1 h of collection) in the laboratory. Untreated feces were placed in three common fecal preservation buffers: RNAlater (Ambion, Austin, TX, USA), 70% ethanol, or 50:50 glycerol:PBS. The fecal sample was homogenized, divided into ninety-six 0.25 g aliquots (four treatment groups, three temperatures, four time points, in duplicate), and placed in 1.5 mL Eppendorf tubes with preservation buffers. All tubes were vortexed to allow buffer penetration in each fecal sample and incubated for two hours at room temperature. Each treatment group was subjected to various temperature conditions: room temperature ($25\,^{\circ}$C), refrigeration ($4\,^{\circ}$C), and freezing ($-80\,^{\circ}$C) on the day of collection (Day 0) and on Days 7, 14, and 56 post-collection. Temperatures were consistently maintained using a microbiological incubator at $25\,^{\circ}$C, a refrigerator at $4\,^{\circ}$C, and a Revco freezer at $-80\,^{\circ}$C. Day 0 samples were processed for DNA extraction after 2 h of temperature treatment, while other samples were stored at the indicated temperatures until DNA extractions on Day 7, 14, and 56.

### DNA extraction

Genomic DNA was extracted from all samples using the 100-prep MoBio PowerSoil DNA Isolation kit (MoBio, Carlsbad, CA, USA). Fecal material was isolated from preservation buffer by pelleting (centrifugation at 10,000x g for 5 min, pouring off supernatant). Samples were placed in bead tubes containing C1 solution and incubated at $65\,^{\circ}$C for 10 min, followed by 1 min of bead beating with the MoBio vortex adapter. The remaining extraction protocol was performed as directed by the manufacturer. DNA concentration was recorded using a QUBIT$^{\text{TM}}$ dsDNA HS Assay and the DNA purity (A260/A280 ratio) was analyzed using a Nanodrop 1000 spectrophotometer (ThermoFisher Scientific, Wilmington, DE, USA).

### PCR and 16S rRNA Sequencing

Bacterial diversity was characterized via amplification by a PCR enrichment of the 16S rRNA gene (V4 region) using primers 515F and 806R, modified by addition of Illumina adaptor and an in-house barcode system (*Lang, Eisen & Zivkovic, 2014*). After an initial denaturation step at $94\,^{\circ}$C for 3 min, we ran 35 cycles of the following PCR protocol: $94\,^{\circ}$C for 45 s, $50\,^{\circ}$C for 60 s and $72\,^{\circ}$C for 90 s, followed by a final hold at $4\,^{\circ}$C. Prior to sequencing, the amount of input DNA per sample was normalized using a SequalPrep Normalization Plate, following the standard protocol (ThermoFisher Scientific, Wilmington, DE, USA). Libraries were sequenced using an Illumina MiSeq system, generating 250 bp paired-end amplicon reads.
## Data analysis and statistics

We used a custom script (available in a GitHub repository https://github.com/gjospin/scripts/blob/master/Demul_trim_prep.pl), to assign each pair of reads to their respective samples when parsing the raw data. This script allows for one base pair difference per barcode. The paired reads were then aligned and a consensus was computed using FLASH (*Magoč & Salzberg, 2011*) with maximum overlap of 120 bp and a minimum overlap of 70 bp (other parameters were left as default). The custom script automatically demultiplexes the data into fastq files, executes FLASH, and parses its results to reformat the sequences with appropriate naming conventions for Quantitative Insights into Microbial Ecology {QIIME v.1.9.1, *Caporaso et al., 2010*} in fasta format. Each sample was characterized for taxonomic composition (number and abundance) using QIIME. For presence/absence analyses, representative operational taxonomic units (OTUs) were clustered at the >97 percent identity level and an OTU table was constructed using QIIME's pick_otus_through_otu_table.py script. In addition, we removed chimeras from the OTU table and filtered for chloroplast and mitochondrial DNA. The resulting table was rarefied at 4000 reads and filtered to remove low abundance OTUs. Raw sequencing data files have been uploaded to NCBI Bioproject (Accession: PRJNA414515) and Figshare at the following link: https://figshare.com/articles/Canine_fecal_samples/5510422.

We compared alpha diversity (mean species diversity per treatment) using the Shannon Index as implemented in the vegan library (*Solymos, Stevens & Wagner, 2016*) in R (*R Core Team, 2016*). We compared OTU richness (number of OTUs found in each sample) and Pielou's evenness (calculated by dividing the Shannon index for diversity by the log of OTU richness). We tested for statistical significance in alpha diversity measures using the Analysis of Variance (ANOVA) with post-hoc Tukey HSD to determine the effects of temperature, storage buffer, and duration of storage in R. We compared beta diversity (the ratio between regional and local species diversity) using Bray–Curtis dissimilarity and weighted Unifrac, and we used PCoA for ordination and clustering. We then used adonis, a multivariate ANOVA based on dissimilarities to test for significant categorical differences with 1000 permutations in the picante library (*Kembel et al., 2010*) in R. OTU frequencies across buffer, temperature, and time were compared using QIIME script group_significance.py. Spearman correlation coefficients and regressions were calculated on R and GraphPad Prism Software.

# RESULTS

To explore the effects of preservation buffer and temperature on the composition, abundance, and quality of bacterial DNA in fecal samples, we performed a longitudinal study over 56 days to evaluate the consequences of different storage methods in grouped samples (Fig. 1).

## DNA concentration and purity

We used ANOVA to test for an effect of storage method on DNA concentration. We detected an overall effect of buffer on DNA concentration ($F$-value = 70.733, $DF = 3$, $P < 0.00001$). In all storage methods, DNA concentration decreased over time

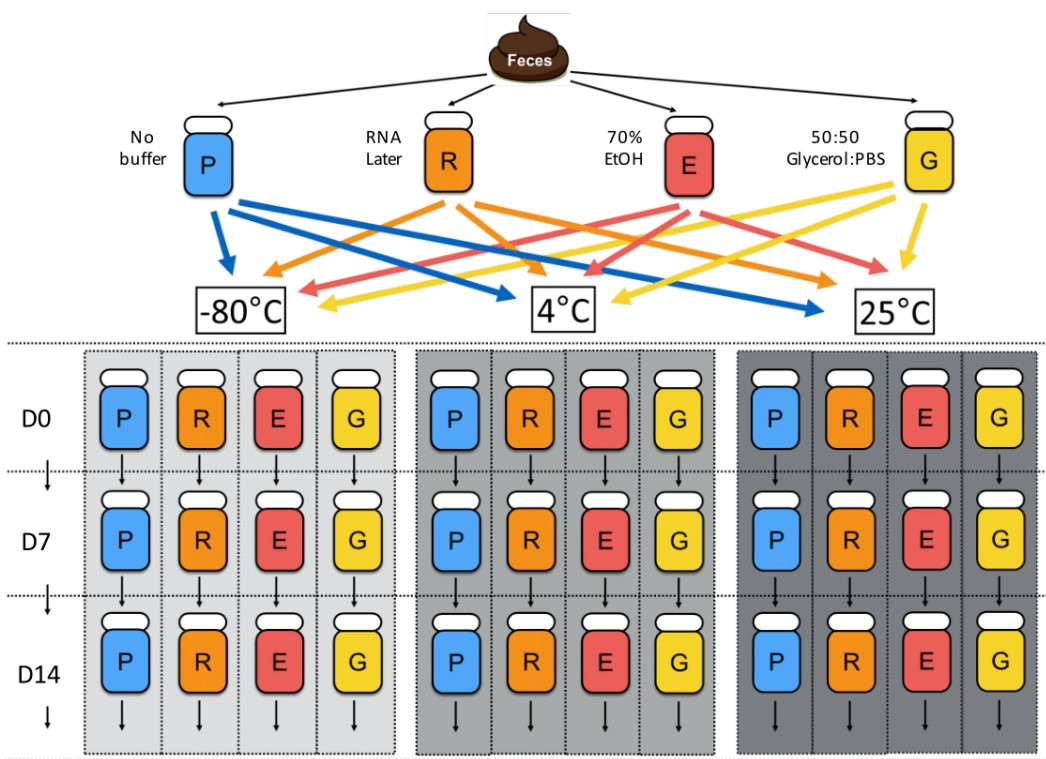

**Figure 1  Chronological flowchart of fecal DNA preservation experiment.** Fecal samples from one canine donor were collected, aliquoted, and treated with no buffer (P), RNA later (R), 70% ethanol (E), and 50:50 Glycerol:PBS (G). After 2 h of incubation at 25 °C, Day 0 samples were immediately processed for DNA extraction, while other samples were stored at the indicated temperatures (room temp: 25 °C, refrigeration: 4 °C, freezing: −80 °C) until extractions on Days 7, 14, and 56.

($F$-value = 7.5, DF = 1, $P < 0.01$). Significant interactions between buffer and time ($F$-value = 3.77, DF = 3, $P = 0.015$) and temperature and time ($F$-value = 9.8, DF = 1, $P = 0.0027$) were also detected.

### By time

The DNA concentration of fecal samples incubated at room temperature without preservation buffer exhibited an 84% decrease in DNA concentration after Day 0 (1/Slope = −0.6812, $R^2 = 0.9935$, $P = 0.0002$) (Fig. 2A), where levels remained through Day 56. We used ANOVA to test for an effect of storage method on DNA quality (as measured by the ratio of A260/A280). We did not detect an overall effect of storage buffer ($P = 0.12$) or storage temperature ($P = 0.66$) on DNA quality. However, length of storage significantly affected DNA quality ($F$-value = 4.365, DF = 1, $P = 0.04$) and there was a significant interaction between storage buffer and length of storage ($F$-value = 2.88, DF = 3, $P = 0.043$). DNA quality declined slightly by Day 14 (1/Slope = −43.90, $R^2 = 0.3135$, $P = 0.0024$) and then increased in the samples stored in 70% ethanol and no storage buffer on Day 56 (1/Slope = 109.4, $R^2 = 0.4400$, $P = 0.0014$) (Fig. S1 ).

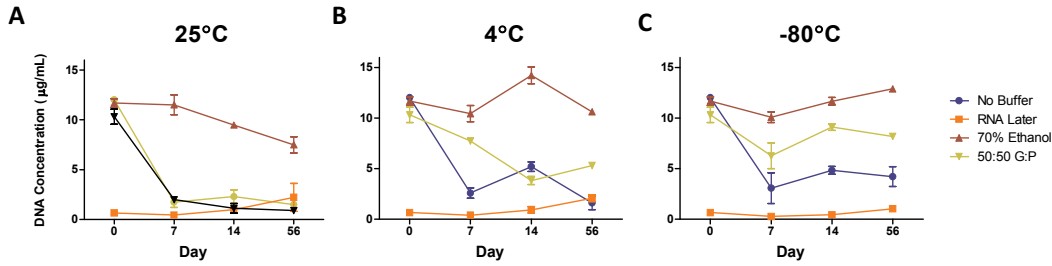

**Figure 2  Average DNA concentration (μg/mL) ± standard error by preservation method over 56 days.** DNA concentrations of samples stored at (A) 25 °C, (B) 4 °C, and (C) −80 °C. Symbols represent average DNA concentrations by buffer and error bars represent standard error of arithmetic means.

### By temperature

Reducing the temperature used to store fecal samples reduced the amount of DNA loss after Day 0 ($1/\text{Slope} = -31.73$, $R^2 = 0.09954$, $P = .0190$). Unbuffered samples and samples in 50:50 glycerol:PBS were most affected by storage at room temperature, while samples in RNAlater and 70/% ethanol were consistent across all temperatures (Figs. 2A–2C). Unbuffered, refrigerated samples exhibited a 1.5-fold higher DNA concentration at Day 7 and 2.25-fold at Day 14 compared to respective samples stored at room temperature. Refrigeration of samples stored in glycerol:PBS samples exhibited a similar trend in DNA concentration with fold-changes up to 4.9 times that of respective samples stored at room temperature, near levels found in fresh control samples. Of the three temperatures observed, freezing at −80 °C yielded the highest DNA concentration over 56 days (AUC = 76.291), with refrigeration close behind (AUC = 72.528), and room temperature with the lowest yield (AUC=52.700). Freezing glycerol:PBS samples at Days 14 and 56 led to a 706% and 811% higher DNA concentration, respectively, compared to samples stored in glycerol:PBS at room temperature. Little change was observed in DNA concentration of samples stored in RNAlater and 70% ethanol. DNA purity remained unchanged across all temperatures ($1/\text{Slope} = -727.4$, $R^2 = 0.0436$, $P = 0.0976$) (Fig. S1 ).

### By buffer

Preservation buffers were evaluated (RNAlater, 70% ethanol, and 50:50 glycerol:PBS) in comparison with unbuffered controls. DNA concentrations in unbuffered samples decreased by 75–80% over time, which were greatly improved with the addition of 70% ethanol regardless of temperature (Figs. 2A–2C). 70% ethanol was the optimal method, exhibiting no significant changes in 56 days ($1/\text{Slope} = -61.48$, $R^2 = 0.0365$, $P = 0.4473$). Preservation with 50:50 glycerol:PBS also improved DNA yield, but only under refrigeration or freezing conditions. Preservation with RNAlater yielded the lowest amount of DNA across all temperature and preservation buffers.

## Bacterial diversity and composition

Alpha Diversity measures: we performed an ANOVA to test for effects of storage buffer, storage temperature, and duration of sample storage on the Shannon Diversity Index, Species Richness, and Species Evenness. Storage buffer had a statistically significant effect

on Shannon Diversity Index values ($F$-value $= 3.07$, $DF = 3$, $P = 0.03$) (Fig. S2 ). Preservation with glycerol:PBS had the highest Shannon Diversity values compared to 70% ethanol ($P = 0.0167$), RNAlater ($P < 0.0001$), and no buffer ($P = 0.0004$). Samples stored with ethanol and without buffer did not differ significantly ($P = 0.5178$), while samples stored in RNAlater had the lowest Shannon Diversity values (Table S1 ). We did not detect an effect of storage temperature ($F$-value $= 2.2$, $DF = 1$, $P = 0.14$) or duration of sample storage ($F$-value $= 0.69$, $DF = 1$, $P = 0.4$) on Shannon Diversity values. Additionally, storage buffer ($F$-value $= 12.4$, $DF = 3$, $P < 0.00001$), duration of sample storage ($F$-value $= 10.8$, $DF = 1$, $P = 0.0016$), the interaction between storage buffer and temperature ($F$-value $= 3.443$, $DF = 3$, $P = 0.0218$), and the interaction between storage buffer and duration of sample storage ($F$-value $= 9.67$, $DF = 3$, $P < 0.00001$) significantly affected observed species richness levels (Fig. S3 ). Effect of storage buffer on species richness levels were similar to that of Shannon Diversity, with the highest in glycerol:PBS and lowest in RNA later. Samples exhibited a reduction in species richness levels at seven days compared to fresh samples ($P = 0.0724$), and a subsequent increase at Day 14 ($P = 0.0018$) and Day 56 ($P < 0.0001$) compared to Day 7. Effects of the interaction between buffer and duration of storage on species richness were also noted (Table S2 ). Lastly, we found no significant effects of buffer, duration of storage, nor temperature on species evenness (Fig. S4 ). Significant effects were associated with interactions between storage buffer and storage temperature ($F$-value $= 3.98$, $DF = 3$, $P = 0.01$), storage buffer and duration of sample storage ($F$-value $= 4.9$, $DF = 3$, $P = 0.004$), and buffer, storage temperature, and duration of sample storage ($F$-value $= 3.1$, $DF = 3$, $P = 0.03$) (Table S3).

Beta Diversity measures: we used a permutational ANOVA to test for effects of storage method on the degree of clustering of bacterial communities in our stored samples. We detected overall effects on bacterial community of storage buffer ($F$-value $= 6.87$, $DF = 3$, $P < 0.001$), storage temperature ($F$-value $= 1.77$, $DF = 3$, $P = 0.037$), and duration of sample storage ($F$-value $= 3.68$, $DF = 3$, $P < 0.001$). Significance testing on the Bray Curtis dissimilarities using a two-sided Student's two-sample $t$-test indicated that samples stored in 70% ethanol and samples stored in glycerol:PBS had lower dissimilarities from fresh samples when compared to samples stored in RNALater and samples stored without a storage buffer (Fig. S5, Table S4 ).

### By time

Complementary to the differences detected in DNA concentration, we found that the bacterial composition in unbuffered samples also changed over time. Visualization of weighted (Figs. 3A–3C) and unweighted (Figs. 3D–3F) PCoA plots showed that while most Day 0 samples (in red) clustered together, some differences were observed between buffers, likely due to immediate shifts in bacterial composition after contact with preservation buffer (Fig. S5 ). From Day 7 to Day 56, unbuffered samples did not cluster with fresh control samples, indicating a shift in bacterial composition between zero and seven days after fecal collection (Figs. 3A–3C). Accordingly, taxonomic evaluation also revealed shifts in relative abundance of bacteria in unbuffered samples, particularly after Day 0 at room temperature. This was exhibited by a marked increase in *Streptococcus*,

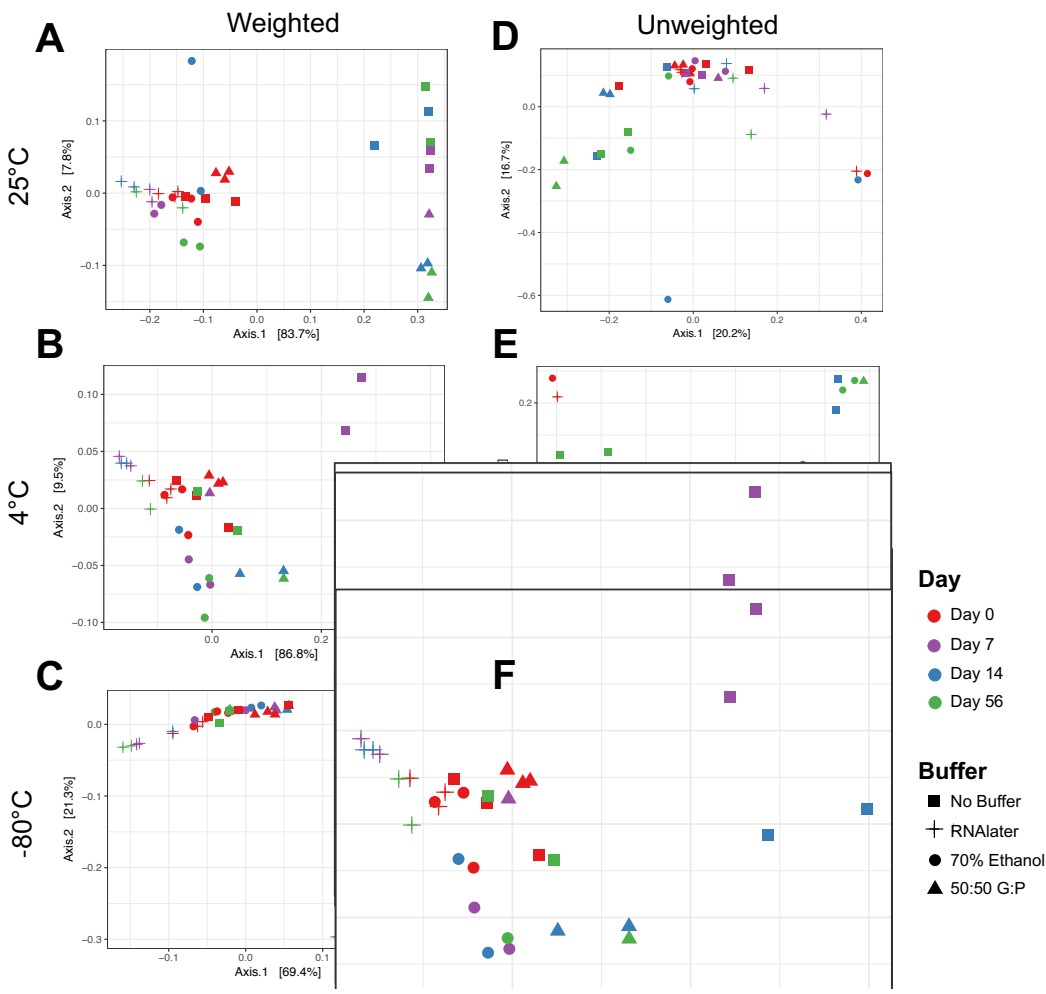

**Figure 3** **Principal coordinate analysis (PCoA) of weighted and unweighted UniFrac distances of microbial communities by temperature.** Weighted Unifrac distances (left) measures relatedness between communities and relative abundance of bacterial phyla at (A) 25 °C, (B) 4 °C, and (C) −80 °C. Unweighted UniFrac distances (right) represent only distances between communities and the evolutionary relatedness of taxa at (D) 25 °C, (E) 4 °C, and (F) −80 °C. Symbols denote method of buffer preservation, while colors represent time over 56 days.

decrease in *Prevotella* (Fig. 4A), and increase in overall Firmicutes:Bacteroidetes ratio (1/Slope $= 1.385$, $R^2 = 0.4687$, $P = 0.0419$) (Fig. 5A) compared to fresh samples. The increase in Firmicutes:Bacteroidetes ratio in unbuffered samples over time was attenuated by temperature reduction (Figs. 5B, 5C). Overall, alterations in OTU frequencies were observed in genera *Collinsella* ($P = 0.0020$), *Enterococcus* ($P = 0.0048$), *Prevotella* ($P = 0.0052$), *Megamonas* ($P = 0.0129$), and *Streptococcus* ($P = 0.0413$) over 56 days.

### By temperature

Weighted PCoA analysis of unbuffered samples indicated that shifts in bacterial composition at room temperature can be minimized by lowering the temperature (Figs. 3A–3C). Samples clustered very closely when stored at −80 °C (Fig. 3C), and segregated with

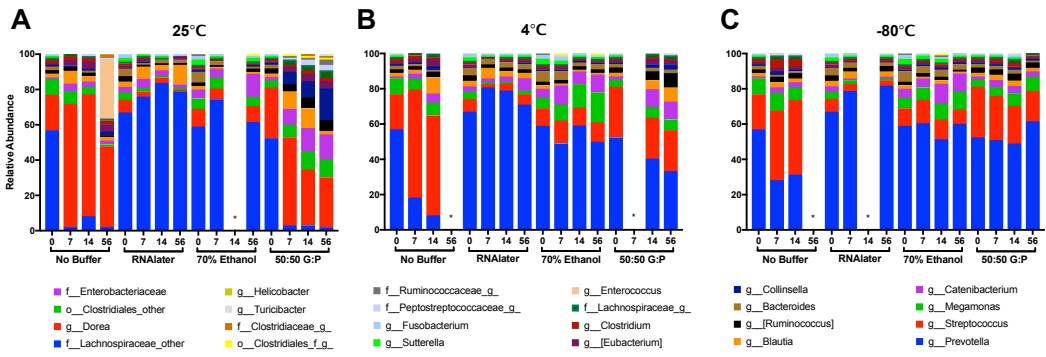

**Figure 4** **Relative abundances of 24 highest taxonomic classifications of bacteria over 56 days.** Each column represents the average of samples collected on Day 0, 7, 14, and 56 at (A) 25 °C, (B) 4 °C, and (C) −80 °C. Canine fecal microbiota was largely dominated by genera *Prevotella* and *Streptococccus*. Comparison of Day 0 samples in different preservation buffers revealed rapid changes in abundance of *Prevotella* and *Streptococcus* with 70% ethanol and RNAlater. *Time points were excluded from samples that did not pass sequencing quality or low abundance OTU filtering.

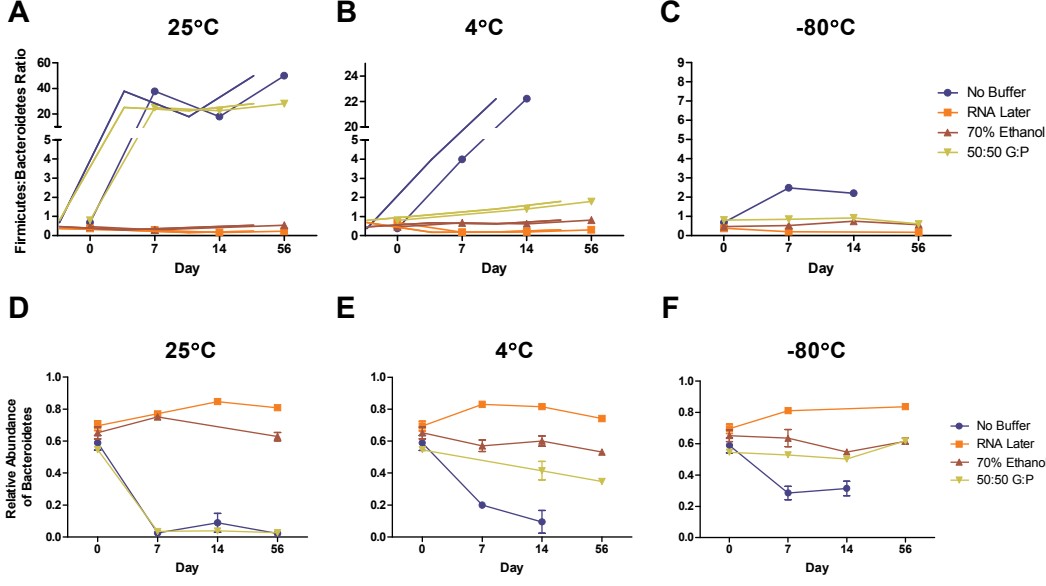

**Figure 5** **Effects of buffer and temperature on relative abundance of bacterial phyla over 56 days.** Symbols represent relative abundances ± standard error of the Firmicutes: Bacteroidetes ratio in samples stored at (A) 25 °C, (B) 4 °C, and (C) −80 °C. Relative abundances of phyla Bacteroidetes in various storage buffers are shown after storage at (D) 25 °C, (E) 4 °C, and (F) −80 °C. Preservation of bacterial composition at the phyla level was optimal under freezing conditions and storage with RNA later or 70% ethanol.

increasing temperature (Figs. 3A, 3B). Samples stored at room temperature without buffer or with 50:50 glycerol:PBS past Day 7 did not cluster with the sample controls at Day 0 (Figs. 3A). Analysis of Bray–Curtis dissimilarity revealed that samples with 50:50 glycerol:PBS clustered closer to control samples with refrigeration and freezing

(Fig. S5 ). However, this pattern was not observed in unbuffered samples. Unbuffered samples did not cluster closely with control samples under refrigeration and freezing conditions. Investigation of the bacterial phyla in these communities revealed a notable reduction in relative abundance of *Prevotella* and increase in *Streptococcus* in unbuffered samples stored at refrigeration (Fig. 4B) and freezing conditions (Fig. 4C). Although freezing is one of the most commonly used tools for stool preservation, we observed that samples frozen without buffer exhibited a lower relative abundance of Bacteroidetes (Fig. 5F) as previously described (*Bahl, Bergstrom & Licht, 2012*) and an increase in Firmicutes:Bacteroidetes ratio (Fig. 5C) compared to samples stored with buffer. Although these alterations were observed in frozen samples without buffer, overall changes in Bacteroidetes and Bacteroidetes:Firmicutes ratios were attenuated by freezing when compared to storage at room temperature (Figs. 5A, 5D or refrigeration (Figs. 5B, 5E). Across all buffers, storage at various temperatures introduced changes in OTU frequencies of genera *Prevotella* ($P = 0.0006$), *Collinsella* ($P = 0.0068$), *Streptococcus* ($P = 0.0094$), *Megamonas* ($P = 0.0094$), *Bacteroidetes* ($P = 0.0094$), and *Catenibacteirum* ($P = 0.0156$).

### By buffer

Weighted PCoA analysis shows that across all temperatures, bacterial communities in samples stored in 70% ethanol clustered relatively closely with those in fresh control samples (Figs. 3A–3C). Samples without buffer did not cluster with fresh samples over time despite temperature reduction. This was consistent with taxonomic evaluation of bacterial genera, as indicated by a large increase in *Streptococcus* and reduction in *Prevotella* (Figs. 4A–4C). Samples stored in 50:50 glycerol:PBS clustered with control samples only under refrigeration and freezing conditions, which was consistent with taxonomic analysis at the OTU level. At room temperature, storage in glycerol:PBS introduced an increase in genus *Collinsella*. Storage with both RNAlater and 70% ethanol rapidly introduced permanent changes in bacterial composition after a 2-hour incubation period on Day 0. RNAlater preservation was consistent across all temperatures, but led to a reduction in relative abundance of *Streptococcus* and *Megamonas*, and increase in *Prevotella* species. Storage in 70% ethanol also preserved fecal microbiota composition at all temperatures, with slight increases in *Catenibacterium* and *Bacteroidetes*. Overall, variation in storage buffer introduced significant changes in OTU frequencies of genera *Streptococcus* ($P < 0.0001$), *Megamonas* ($P < 0.0001$), *Collinsella* ($P < 0.0001$), *Catenibacterium* ($P < 0.0001$). Although statistically insignificant, alterations in genera *Prevotella* ($P = 0.0563$) and *Bacteroides* ($P = 0.0758$) were also noted across storage buffers.

## DISCUSSION

There is increasing evidence highlighting the importance of bacterial DNA preservation in multitude of settings, including health evaluations, research endeavors, and forensic science. Characteristic signatures of microbiota have been explored as a result of the availability of next-generation sequencing, extending our knowledge beyond culturable methods. One of the most readily available resources to study microbes in humans and animals is fecal collection (*Hale et al., 2016*). Not surprisingly, microbiota, including

that of the gut, is often transient and dynamic, posing a challenge for scientists to make sense of samples post-collection. Bacterial DNA can be degraded through environmental perturbation and subsequent hydrolysis, oxidation, and methylation, supporting the need to limit spontaneous decay (*Lindahl, 1993*). Nevertheless, there remains considerable value in distinguishing changes in the microbiome in high resolution for improving animal, human, and environmental health. This is accompanied by a need for more effective storage methods that precisely and accurately capture the bacterial community at a given time-point. Additionally, studying longitudinal changes in microbiota has paved the way for developing microbiome tools to study unique signatures and providing, at least in theory, the means to locate people in space and time in forensic science (*Kim, Zorraquino & Tagkopoulous, 2015*; *Fornaciari, 2017*). A subset of our study investigates longitudinal changes in fecal microbiota at various temperatures, which provides insight on how various bacterial signatures could be maintained or altered over time. Studies tracking how certain microbes change over time could be utilized to interpret chronological events in the past. While this study focuses on optimal preservation of DNA for future analysis, information gathered at various time points is valuable to extrapolate longitudinal data in a forensic context. It is evident that investigation of optimal methods for DNA preservation will have important impacts on microbiome studies in field, clinic, and laboratory settings. Assessing changes in preserved fecal microbiota over time provides insight on whether changes we observe are biologically relevant and useful for outcome measures.

A variety of studies have examined preservation methods to minimize post-sampling alterations in fecal bacterial DNA. Such experiments have evaluated fixation conditions with 95% ethanol, 70% ethanol, FTA card, OMNI gene Gut, RNAlater, glycerol, refrigeration, and freezing. However, there is little consensus on the optimal buffer and temperature condition for bacterial preservation (*Nechvatal et al., 2008*; *Cardona et al., 2012*; *Kolodziej et al., 2013*; *McKain et al., 2013*; *Fliegerova et al., 2014*; *Hale et al., 2015*; *Song et al., 2016*; *Hale et al., 2016*; *Metzler-Zebeli et al., 2016*). Furthermore, a limited number of studies have evaluated the combination of these chemical buffers with temperature reduction to optimize DNA preservation. Our study examined the influence of three commonly used preservation buffers on the bacterial integrity of canine feces after eight weeks of storage at room temperature (25 °C), refrigeration (4 °C), and freezing (−80 °C). Since we were interested in measuring change over time in identical fecal samples, we utilized one canine donor and homogenized the stool sample prior to tube allocation. While one donor was used for this study, we speculate that there may be inter-individual differences in storage because different dogs may have different bacterial compositions and some fecal bacterial groups may be more prone to temperature or buffer alterations. Therefore, more studies involving additional animals and species are needed to make definitive conclusions about bacterial changes with preservation methods. We report that fecal DNA concentration and microbial composition changes greatly over time, and that the common practice of preservation by freezing may not be adequate in maintaining bacterial DNA. Overall, total DNA recovery and fecal composition of samples stored at 4 °C were similar to that of samples stored at −80 °C over 56 days, both of which were vastly different from samples stored at 25 °C. Minor differences between storage at 4 °C and −80 °C were observed

in samples without buffer, which exhibited higher abundances of *Streptococcus* and lower abundances of *Prevotella* at 4 °C compared to −80 °C. Storage in buffers such as 70% ethanol, RNA later, and 50:50 glycerol:PBS greatly reduced the changes observed between samples stored at 4 °C and −80 °C. This suggests that deep freezing may not be needed when samples are stored with a preservation buffer, a particularly useful tool in field conditions or settings without a laboratory-grade freezer.

To evaluate methods that are most cost-effective and accessible to all studies, we were especially interested in the efficacy of ethanol preservation. Previous studies have utilized 70% and 95% ethanol, each with varying results in DNA yield and microbial community stability (*Hale et al., 2015*; *Song et al., 2016*). Efficacy of ethanol preservation may be dependent on concentration due to species-species differences in stool consistency. Based on the lack of moisture in our canine stool sample, we decided to use 70% ethanol for fecal preservation. We found that it yielded the highest amount of DNA and most closely resembled that of fresh samples within 56 days compared to other buffer solutions. Song, et al. reported that 70% was inadequate for DNA preservation, but sample preparation methods did not include a homogenization step after the fecal sample is immersed in ethanol. The penetration of 70% ethanol in fecal samples, along with stool consistency, may play a role in DNA preservation, which we attempted to circumvent by including a homogenization step. Additional experiments are warranted to examine 70% and 95% ethanol using a homogenizing protocol and various stool consistencies. We also evaluated DNA preservation by RNAlater, a historically-supported and commonly used DNA stabilization buffer (*Schnecker et al., 2012*). Consistent with our findings, studies have recently shown that this method yields very low amounts of DNA (*Hale et al., 2015*; *Song et al., 2016*) due to degradation. We hypothesize that residual RNAlater remaining in fecal samples may interfere with the cell lysis and protein digestion in DNA extraction, inhibiting optimal DNA isolation. Investigation of DNA concentration using both a QUBIT$^{TM}$ dsDNA HS Assay and a Nanodrop 1000 spectrophotometer revealed that Nanodrop-obtained values were markedly higher in RNAlater samples than that of QUBIT (Fig. S6 ). Due to the methodological variability of DNA concentration in samples with RNAlater, deeper analyses of bacterial composition was performed. Our data showed that both RNAlater and 70% ethanol were effective and consistent across all temperatures, and that DNA yields did not reflect bacterial composition. While preservation buffers are a useful tool, it is important to note that storage in both RNAlater and 70% ethanol rapidly introduced changes in bacterial composition, even at Day 0. Compared to fresh, unbuffered samples, ethanol preservation slightly increased the relative abundance of *Catenibacterium* and *Bacteroidetes* while RNAlater preservation greatly increased abundances of *Prevotella* and decreased *Streptococcus* species. These alterations did not change over time.

It is clear that post-sampling conditions may impact results and interpretations of bacterial DNA. Bacterial DNA outcomes are widely used in medicine and healthcare, such as the Firmicutes:Bacteroidetes ratio, a comparison of butyrate to propionate/acetate-producing bacteria (*Bahl, Bergstrom & Licht, 2012*). This ratio, along with levels of Proteobacteria and Actinobacteria, is used as a potential indicator of gastrointestinal health and immune balance (*Honnefer, Minamoto & Suchodolski, 2014*). While there is

no direct comparison to disease state in this study, we showed that alterations in fecal DNA composition can markedly shift this ratio, particularly with inadequate temperature reduction or preservation buffer. More studies investigating fecal DNA preservation and stability in chronic diseases are warranted. Preservation buffers such as glycerol at room temperature may promote the selective growth of bacteria such as *Collinsella*, which has been linked to production of inflammatory cytokine IL-17A and disease states such as rheumatoid arthritis (*Chen et al., 2016*). Use of carbon-containing buffers like glycerol may provide energy sources for certain bacterial groups, creating disease preservation biases that could be falsely interpreted as a clinical concern (*Murarka et al., 2008*). Therefore, careful selection of preservation buffers is advised when measuring specific biological outcomes. In this study, we also showed that despite temperature reduction, bacterial DNA changes over time without buffer, highlighting the need to consider post-collection dynamics in microbiome research. Changes in bacterial composition observed in unbuffered samples were eliminated with the addition of preservation buffers such as 70% ethanol, RNA later, and 50:50 glycerol:PBS. For example, the use of 50:50 glycerol:PBS in refrigeration and freezing temperatures improved both DNA composition and concentration compared to unbuffered controls at room temperature. While temperature reduction preserved DNA to an extent, storage buffer had the greatest impact on DNA preservation. Storage of fecal DNA in 70% ethanol was the optimal preservation method across all temperatures, highlighting its utility in settings without access to temperature control. Our data suggests that caution should be taken in sample handling and use of adequate storage buffers to accurately and consistently analyze fecal microbial DNA.

## CONCLUSIONS

A 56-day longitudinal study of fecal microbiota from one canine donor was conducted to evaluate storage conditions with RNAlater, 70% ethanol, 50:50 glycerol:PBS, and no buffer at −80 °C, 4 °C, and 25 °C. We report that temperature, time, and buffer significantly changed the composition of fecal microbiota, which was comprised predominantly of genera *Streptococcus*, *Prevotella*, *Collinsella*, and *Megamonas*. Samples stored without buffer exhibited DNA degradation and altered composition and diversity despite temperature reduction, suggesting that gold standard methods of immediate freezing at −80 °C may not be optimal for fecal preservation. The efficacy of DNA preservation was largely driven by storage buffer, which produced an additive effect in glycerol:PBS when combined with temperature reduction. Most notably, fecal preservation with 70% ethanol yielded DNA concentrations and bacterial composition closest to that of fresh samples at all temperatures, highlighting the potential of its utility in field, laboratory, and clinical settings without access to a laboratory freezer. Because this study utilizes samples from only one canine donor, research involving more animals and species are warranted to better evaluate the efficacy of fecal microbiota preservation. In conclusion, this study underlines the need for more comprehensive evaluation of fecal DNA storage methods for accurate downstream microbial analysis.

## ACKNOWLEDGEMENTS

The authors would like to acknowledge Dr. Jan S. Suchodolski, Gastrointestinal Laboratory, College of Veterinary Medicine and Biomedical Sciences, Texas A&M University, for his critical review of our manuscript. We thank members of the Eisen lab for technical assistance at various time points in the study, and Clarissa Santos Rocha for providing training and expertise on QIIME and R software.

### Funding

The project described was supported by Nestle Purina PetCare Company (St. Louis, MO) through grant number OVCR-201503617 and the Center for Companion Animal Health (CCAH) at the University of California, Davis, School of Veterinary Medicine through a matching fund award. Financial support was provided by the Students Training in Advanced Research (STAR) Program at UC Davis through Boehringer Ingelheim (Merial). The funders had no role in study design, data collection and analysis, decision to publish, or preparation of the manuscript.

### Grant Disclosures

The following grant information was disclosed by the authors:
Nestle Purina PetCare Company: OVCR-201503617.
Center for Companion Animal Health (CCAH).
School of Veterinary Medicine.
Students Training in Advanced Research (STAR) Program.

### Competing Interests

Jonathan A. Eisen is an Academic Editor for PeerJ. Holly H. Ganz is the CEO of AnimalBiome, Inc., a company developed to improve gut health in companion animals.

### Author Contributions

- Katti R. Horng conceived and designed the experiments, performed the experiments, analyzed the data, contributed reagents/materials/analysis tools, prepared figures and/or tables, authored or reviewed drafts of the paper, approved the final draft.
- Holly H. Ganz conceived and designed the experiments, performed the experiments, analyzed the data, contributed reagents/materials/analysis tools, authored or reviewed drafts of the paper, approved the final draft.
- Jonathan A. Eisen and Stanley L. Marks conceived and designed the experiments, contributed reagents/materials/analysis tools, authored or reviewed drafts of the paper, approved the final draft.

## Data Availability

The metagenome data can be accessed through NCBI BioProject ID PRJNA414515. (https://www.ncbi.nlm.nih.gov/bioproject/PRJNA414515).

Jospin, Guillaume (2017): Canine fecal samples. figshare. Dataset. Available at https://doi.org/10.6084/m9.figshare.5510422.v1.

## Supplemental Information

Supplemental information for this article can be found online at http://dx.doi.org/10.7717/peerj.4827#supplemental-information.

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
