# Peer review of "Effects of preservation method on canine (Canis lupus familiaris) fecal microbiota"

_PeerJ, doi:10.7717/peerj.4827_

## Round 0.1 · original submission · Major Revisions

As you will see, Reviewer 1 in particular provides a large number of constructive suggestions regarding further improvement of your manuscript, especially regarding data analysis and interpretation. Most importantly, probably, are issues with statistical analyses; focus on higher taxonomic levels; and the fact that only one individual was sampled. I agree with reviewer 2 that this approach, while allowing to assess the impact of various technical factors, neglects the fact that individual microbiomes might provide different challenges in terms of technical biases, and thus, individual variation should at least be addressed.

·

Basic reporting

Areas for improvement
• Are the raw sequencing data files uploaded to a publicly available database? I did not see this noted in the methods.
• Fig. 3 – need buffer symbols in the weighted diagrams.
• 96 samples were prepped, but the PCoAs only show ~34 samples - what happened to the other samples?

Minor concerns
• Methods: Did the donor have any recent antibiotic exposure? (This information isn’t critical to you study, but it would be worth mentioning this history in the methods.)
• Need a period after the reference in line 120, 293, 297 and line 368 and 377 – did you intend to have a line break here? (Please check carefully throughout the paper to ensure appropriate punctuation.)
• Fig. 2, S1, 5A,B – mention if the error bars SD or SE in the legend.
• Fig. 2 and 5 are a little challenging to read because the location of the error bars distort the shapes, so it becomes difficult to tell the circles / triangles / squares apart. If you are considering publishing in color, the use of color in these graphs may be helpful. Otherwise, you could also consider adding varying dashed lines between points to make it easier to tell each method apart.
• Line 289 – “Assessing changes in preserved fecal microbiota over time provides insight on…” – suggest adding the word “preserved” to this sentence to clarify. Can also remove the word “would” and use “provides” instead of “provide”

Experimental design

Currently the authors provide statistical results for DNA concentration and purity, alpha diversity, and PERMANOVA results indicating that there are “effects on bacterial community of storage buffer, temperature, and duration”; however, these effects are never explored statistically. Instead, the authors focus on high level (phyla/class) taxonomy summaries and PCoA visualizations. This study would be greatly improved by further statistical analysis at the genus / OTU level to answer questions like – are there any microbes/OTUs that differ significantly between preservation treatments? Do any of the methods bias / enrich the preservation of particular microbes / OTUs? (This can be accomplished in QIIME: group_significance.py and supervised_learning.py.)

Critical areas for improvement:
• Lines 206-218: There are no figures or detailed results provided for the alpha diversity results. The results state that buffer, storage duration, and temperature had statistically significant effects on the various alpha diversity metrics (Shannon, observed species richness, evenness) - but do not indicate which buffer / duration / temperature had higher/lower diversity metrics. This is important information to share either in figures or words so that readers will be aware of how microbial diversity can be affected by storage method.
• My interpretation of Fig 4A,B are that NOT all day 0 samples look the same. Samples preserved in ethanol or RNAlater – even for just 1 hour prior to extraction at Day 0, look different than samples without a buffer or with glycerol:PBS. This is detectable at a phylum level, which means there is a very strong and rapid bias introduced by these preservation methods. In order to understand this better, analysis at the OTU level is required followed by interpretation of these results.
• Fig. 5C reports mean fold change of frozen samples compared to “fresh samples.” Were all Day 0 (“fresh”) samples combined for this analysis? As suggested above, I do not believe that all Day 0 samples are comparable / similar and that RNAlater and Ethanol may have introduced some taxonomic biases into preservation even at Day 0. I suggest removing this analysis. Deeper OTU level analysis will provide more thorough statistical results to explore for changes over time introduced based on buffer.

Minor concerns
• Fig. 6 is not a particularly valuable analysis and can be removed from the paper. The microbial community actually show this same result in much more detail (see Fig 4A), and through taxonomic analysis (at higher taxonomic levels and the OTU level) you can provide a deeper interpretation of these results rather than a correlation. (For example, after appropriate analyses, you may be able to say something like: Day 0 microbial communities all contained Fuso / Proteo but these microbes appeared to degrade in the glycerol / no buffer methods over time (except at -80C) as they were no longer identifiable / significantly decreased from Day 7 – Day 56 at 25C and 4C.) OTU level analyses may identify several other biases in the data by storage method that will be valuable to report.

Validity of the findings

• Line 326-327 – this study in this reference was only done for 72 hours, so it is misleading to suggest (based on this reference) that refrigeration would be adequate storage when your study was for much longer - 56 days. Additionally, in your study, microbial communities were examined at the level of phyla and class – which are very high taxonomic levels. Genus and species differences are potentially much greater over time and by buffer, and these types of changes are likely more important and relevant to any type of health or research study. No data / results are provided at the genus/species taxonomic levels, so again, suggesting that refrigeration is adequate is not appropriate here. (However, if you do perform analyses at the genus or “OTU” level and find no significant differences between time/buffer for refrigeration, then you can certainly make the conclusion that refrigeration of fecal samples over 56 days is reasonable. I suspect this will not be the case.)
• Lines 175-177 – Interesting that DNA quality declined across all preservations methods but only at Day 14 (but not day 56). Potentially suggests an external factor – new/different person did the extraction that day? New/different reagents? Accidental minor modification to extraction protocol?
• Line 292 – “This approach” - what approach? – are you referring to preservation of microbiota? This seems unrelated to the reference in this sentence which is about how microbial gene expression can be used to predict the cellular and environmental states from which a microbe was derived.
• Line 292-300 – It is true that microbial forensics is a growing field both to understand microbial environmental exposures based on gene expression as well as to understand processes like decomposition and historic pandemics; however, why spend a full paragraph on microbial forensics? It is my understanding based on the 3 references included in this paragraph that these techniques focus on examining present microbes in order to interpret past events rather than “preserving” DNA to examine in the future. Either add a clearer explanation of the value of DNA preservation within this forensic context or perhaps choose to highlight other areas in which DNA preservation is critical (e.g. field studies that don’t have immediate access to labs, research/clinical studies in which DNA must be preserved for future sequencing)
• Lines 374-377, providing clinical relevance of Coriobacteriia is not especially helpful when presented in this way. You could link it to your preservation study (e.g. There was a bias toward preserving (or promoting the growth of?) Coriobacteriia in glycerol preservation. This bias is important to know because Corobacteriia has clinical relevance and could falsely be identified as increased if an individual used this preservation method.)
• Lines 378-379 – remove, this seems like a random statement / reference here. (1. Your study focuses on “unbiased” microbial community preservation not selective bacterial enrichment. 2. Your results and conclusions highlight that freezing is more effective than room temp preservation.)
• Lines 388-391 – These 2 sentences seem to contradict each other. The first sentence says there’s not much difference between samples stored at 4 and -80; the second sentences days temperature is important because it has an additive effect combined with buffer. Within that same sentence it is also stated that preservation is really driven by buffer. Suggest rewording for clarity.
• Depending on the OTU-level analysis, it may be important to discuss the possibility that some microbes were enriched at room temperature or 4C. (For example, while your DNA concentration results indicate relative stable DNA concentrations from day 7-56 at the lower temperatures, this may be because some microbes bloomed while others died / DNA degraded. Concentration results should be evaluated within the context of OTU analysis. As a researcher, I don’t just want high DNA concentrations, I want to know that I am accurately preserving a microbial community, so if I get high DNA concentrations but they are concentrations of microbes that bloomed while in the refrigerator – this is not a useful preservation method.

Additional comments

This manuscript describes an important and technically sound study examining various preservation methods (25C, 4C, -80C, RNA later, 70% ethanol, 50:50 glycerol:PBS) for canine stool. As investigations of the microbiome continue to grow in number and value, it will be critical for clinicians and researchers alike to understand how various preservation methods affect their samples and what the best options are for preservation. Additionally, this study introduces the value of combining preservation methods (both temperature and buffers) in order to achieve optimal results. That said, one of the important areas for improvement is a deeper analysis of the microbial community data. The importance of DNA preservation lies in the ability to maintain microbial communities over time that accurately represent the microbial communities present in the non-preserved, immediately extracted sample. If samples are only analyzed at the phylum (or class) level, then there is no way of knowing if specific OTUs were maintained over time or by each buffer – which undermines the value and power of this experimental design.

Thank you for allowing me to review this well-designed study and all the best!

Reviewer 2 ·

Basic reporting

No comment

Experimental design

Fatal flaw: n=1

Validity of the findings

no comment

Additional comments

The authors have described a study evaluating the effects of storage temperature and preservation buffers on fecal microbiota of dogs. This study may have been of interest if done properly. Unfortunately, only 1 animal was sampled so it is unlikely that the results can be applied to any population. An n=1 is not useful. Disease, drugs, diet, etc. can all drastically affect gastrointestinal microbiota populations that will differ greatly in their resistance to temperature, oxygen exposure, freeze-thaw cycles, preservation buffers, etc. Therefore, I find that this is a fatal flaw and not worth reviewing the paper. Other papers with the same goals have sampled multiple humans or animals and I feel the same must be done here to be valuable.

Here are just a couple examples in recent years:
-Gorzelak et al., 2015, PLoS ONE 10:e0134802 (4 people)
-Anderson et al., 2016, Sci. Rep. 6:31731 (16 people)
-Blekhman et al., 2016, Sci. Rep. 6:31519 (13 rhesus macaques)

---

## Round 0.2 · Minor Revisions

First of all, I would like to thank you for a very good job towards improving the manuscript, using suggestions provided.

I fully agree with the reviewer, that your manuscript has very much improved and that there are only minor issues that still need some attention.

Most importantly, those concern the statistical analyses. It would be very interesting, and of practical relevance, if post hoc tests could be added to pinpoint more specifically at e.g. which storage buffer performs better than others, rather than indicating that the factor storage buffer has a significant effect.

Obviously, as samples from different individuals can be characterized by different microbiota, conclusions drawn from your study should ideally be confirmed by tests with additional samples from canines and other host species, but I understand that this is beyond the scope of your study.
It is somewhat curious that your main recommendation regarding the use of 70% ethanol is pointing into the exact opposite direction as the study published in 2016 by Song et al..This contrasting finding is, however, not really discussed in your manuscript.

·

Basic reporting

NA

Experimental design

NA

Validity of the findings

NA

Additional comments

The authors have thoroughly responded to reviewer comments and added several helpful analyses, interpretations, and clarifications to their manuscript. I suggest only the minor revisions listed below.

Line 219: "the interaction between storage buffer and temperature (F-value=3.443, DF=3, P=0.22) […] significantly affected observed species richness levels (Fig. S3)." - The way this sentence is written looks like the buffer/temperature interaction "significantly affected" - but the p value (0.22) is not significant? (Fig S3 indicates this interaction should be p = 0.022)

Line 214-215: "Storage buffer had a statistically significant effect on Shannon Diversity Index values (F-value=3.07, DF=3, P=0.03)." - Results should include post-hoc tests to determine what/which buffers have significantly different Shannon diversity values. (Your readers will be especially interested to know which buffers are most effective at preserving / maintaining a combination of richness/evenness (Shannon).)

Lines 217-221: "the interaction between storage buffer and temperature (F-value=3.443, DF=3, P=0.22), and the interaction between storage buffer and duration of sample storage (F-value=9.67, DF=3, P<0.00001) significantly affected observed species richness levels (Fig. S3)." - Results should ideally include post-hoc tests to determine what combinations within the interactions have significantly different observed species richness levels. This could be in a supplemental table.

Line 221-225: "In addition, for species evenness, we found that there were significant effects associated with interactions between storage buffer and storage temperature (F-value=3.98, DF=3, P=0.01), storage buffer and duration of sample storage (F-value=4.9, DF=3, P=0.004), and buffer, storage temperature, and duration of sample storage (F-value=3.1, DF=3, P=0.03) (Fig. S4)." - Results should ideally include post-hoc tests to determine what combinations within the 3-way interaction have significantly different observed species evenness. This could be in a supplemental table.

Line 241-245: Need to italicize microbial genera. (Also in legend of Fig. 4) - check throughout the rest of the document for this (i.e. line 263, 271-273, 280)

Line 262: Replace "phyla" with "taxa"

Line 270: Fig. 5C is listed in the manuscript but I only see 5A and 5B in the figures.

Line 276: PCoA's are just a visualization (not an "analysis") - unless you ran a statistical analysis (i.e. PERMANOVA) and can list the results? Lines 276-278 state that Weighted PCoA demonstrates that "RNA clustered very closely with those in fresh control samples." When I look at Fig. 3 It looks like RNA samples cluster somewhat separately (on their own) from fresh control samples. (Do you have a statistical analysis based on the distance matrices that indicates that RNA samples are not statistically distinct from fresh control samples?)

Line 234-237 states: "Analysis of weighted and unweighted PCoA plots show that most Day 0 samples (in red) clustered together, indicating a high degree of similarity among samples extracted regardless of buffer preservation at the starting time point." However, line 288-290 states, "Storage with both RNAlater and 70% ethanol rapidly introduced permanent changes in microbial composition after a 2-hour incubation period on Day 0." Consider rewording lines 234-237 for consistency.

Line 440: remove "both" (or move "both" before "effective")

Line 475: Suggest rewording to "creating a preservation bias that could be falsely interpreted as a clinical concern"

Line 547: Suggest removing "comprised of genera Streptococcus, Prevotella, Collinsella, and Megamonas."

---

## Round 0.3 · accepted · Accept

Thanks for turning around your revision so effectively, and further improving your manuscript, which I think will be a very useful addition to this field.